# Bibliometric Analysis of Cross-Sectional Studies on Early Childhood Caries

**DOI:** 10.3390/healthcare13091067

**Published:** 2025-05-06

**Authors:** Rana A. Alamoudi

**Affiliations:** Pediatric Dentistry Department, Faculty of Dentistry, King Abdulaziz University, Jeddah 21589, Saudi Arabia; rasalamoudi@kau.edu.sa

**Keywords:** early childhood caries, bibliometric analysis, VOSviewer, co-authorship, co-citation

## Abstract

Background/Objectives: Early childhood caries (ECC) is a significant global public health issue with economic and psychosocial consequences, impacting families and pediatric dentists. It affects children’s quality of life, causing pain and infection. Despite increasing research on ECC cross-sectional studies worldwide, inconsistencies and gaps remain in terms of geographical disparities. This study aimed to conduct a bibliometric analysis of cross-sectional surveys on ECC by examining the co-authorship, citation analysis, co-citation networks, and keyword co-occurrence. Methods: An advanced search was performed using relevant terms in the Dimensions database from 2005 to 2024. Bibliometric parameters were retrieved through the database’s analytical view tool and VOSviewer software. Results: A total of 571 documents were identified, with the highest output between 2019 and 2023 (355 records). Saul Martins Paiva authored the most articles (10), with 294 citations and a total link strength of 19. Brazil and the U.S. had the highest numbers of publications (56 and 52) and total link strengths, i.e., a measure of collaborative ties (21 and 50). The Universidade Federal de Minas Gerais in Brazil had the most published documents (15). *BMC Oral Health* led in terms of citations (44 articles, 899 citations, average 20.43%). The frequently co-occurring terms included ECC (1147 occurrences), oral health (417), and preschool child (301). Conclusions: This bibliometric analysis highlights the global interest in cross-sectional ECC studies beyond pediatric dentistry, helping researchers understand the field’s scope and progress.

## 1. Introduction

Early childhood caries (ECC) refers to all the carious lesions found in the primary teeth of young children from birth until the age of 71 months [1]. The Global Burden of Disease Study reported that dental caries in primary teeth is the 12th most common condition worldwide, impacting approximately 560 million children. In developing countries like Lao, the Philippines and Cambodia, and in certain groups within developed nations, such as those in rural areas and migrant communities, the prevalence of ECC can be as high as 90%. Although the global occurrence of ECC has declined in recent years, it remains a significant public health issue [2].

ECC is a severe form of dental decay that starts shortly after teeth erupt and advances quickly [3]. This condition has been referred to by various terms, including nursing bottle caries, milk bottle syndrome, rampant caries, nursing caries, baby bottle tooth decay, prolonged nursing habit caries and baby bottle caries [4]. ECC can have a negative impact on a child’s psychological, physical as well as social well-being, as oral pain and tooth loss can impair chewing and speaking abilities, affecting their overall health-related quality of life (OHRQoL) and that of their families. Additionally, treating ECC in very young children can be difficult, often requiring dental procedures under sedation or general anesthesia, which are costly, time consuming, and carry potential risks [5].

This disease is characterized by a multi-factorial etiology, including vulnerable teeth and host, dietary sugars, cariogenic bacteria and time [6]. The origins of this multifaceted issue can be traced to breast or bottle feeding ad libitum, high sugar consumption, inadequate exposure to fluoride, enamel abnormalities, and subpar oral hygiene practices [7]. While preventive measures such as early assessment, identifying individual risk factors, educating and counseling parents/caregivers, and initiating treatments like fluoride topical application can significantly curb the progression of ECC, which generally poses a threat to the overall health and well-being of afflicted infants, toddlers as well as children, neglecting to address and prevent dental issues can lead to serious long-term consequences [8]. Additionally, ECC in the early years has been linked to dental problems later in childhood [9].

Given the negative impacts of ECC on both oral and overall health, extensive research has been conducted and published in academic publications [10,11,12,13]. The body of scientific literature in this arena has been expanding steadily, with a notable increase in both the number of journals and the breadth of content. This proliferation presents a challenge for academicians, emerging researchers and students seeking to identify the most significant contributions to the field. Therefore, it is essential to analyze this expansion to understand its outcomes and influence on research. One effective approach for such an analysis is through bibliometric methods, which involve statistical analysis of the literature to uncover its historical development [14].

Bibliometrics serves as a quantitative approach to scrutinizing the literature within a particular field, like dental caries, offering valuable insights into the frequency, pattern and distribution of research activities. Such studies undertake a range of analyses to present the scientific output on a subject systematically. They also gauge the influence of individual researchers and research groups by leveraging citation metrics like the citation counts, author productivity, journal distribution, and geographic locations and affiliations of research institutions involved [15]. These analyses can pinpoint the most prolific authors, institutions, and countries involved. The data gleaned from bibliometric analyses are instrumental in tracking changes in scientific output trends over time, highlighting the primary areas of research focus and their evolution. Moreover, these analyses offer an overarching view of the existing research and help pinpoint gaps in the research landscape [16].

In recent times, bibliometric research has delved into various facets concerning ECC. Vega et al. conducted a bibliometric analysis focusing on the risk factors associated with ECC [17]. Additionally, Patil et al. scrutinized the 100 most commonly referenced articles on ECC [18], while Melo et al. performed a bibliometric study on research into caries diagnosis [19]. Although these studies have significantly enriched the foundational knowledge on ECC, there is a notable absence of bibliometric studies exploring the scientific output concerning various cross-sectional surveys regarding ECC. Moreover, researchers often find it challenging to swiftly grasp the current advancements, frontiers, and future trajectories in this domain due to the intricacies of the existing literature. Thus, the aim of this study is to conduct a comprehensive bibliometric and visualization analysis of research pertaining to various cross-sectional surveys of ECC. This analysis encompasses publication data, co-authorship patterns, countries involved, organizational affiliations, citation trends, co-citation networks and keyword co-occurrences. By doing so, the aim of this analysis is to furnish a theoretical foundation for future investigations in this field.

## 2. Materials and Methods

### 2.1. Data Source

In this study, I combed through the Dimensions database (Digital Science & Research Solutions Ltd., London, UK) to find pertinent articles. Dimensions stands out as a crucial resource in the realm of interconnected research data, comprising top-tier academic journals known for their quality.

### 2.2. Search Strategy

In the exploration process, I opted for the expression “early AND childhood AND caries AND cross-sectional AND (study OR studies OR survey OR surveys)” within the search engine of the Dimensions database. This search was confined to the “Title and abstract” section. When crafting the search strategy, a timeframe spanning from 2005 to 2024, encompassing roughly 20 years, was selected. The search was conducted on 29 May 2024, and all the findings were gathered on the same day to prevent potential biases stemming from temporal factors. To refine the results, the search was restricted to only “Articles” under the “Publication Type” category, thereby excluding preprints, chapters, and proceedings. Subsequently, upon screening the titles and abstracts, publications deemed irrelevant to this study’s focus were eliminated. Two researchers were tasked with navigating the database and assessing the relevance of the identified publications. Any discrepancies were resolved through discussions involving a third researcher (examiner validation), culminating in a consensus. All the volunteer researchers were acknowledged. Following these outlined criteria, a total of 571 publications were ultimately deemed suitable for inclusion in this study. Comprehensive records and references, encompassing details such as the title, author, country, institution, journal, abstract, keywords and references, were exported. Given the nature of this study as a bibliometric analysis, which did not involve human or animal subjects, ethical approval from an ethics committee was not sought or required.

### 2.3. Data Analysis

The data set generated was saved as a Microsoft Excel file (Microsoft 365, American Multinational Technology Corporation, Redmond, WA, USA) with a CSV file extension. This file was then imported into the VOSviewer software (version 1.6.20, Leiden University, The Netherlands) which is a freely available tool used for visualizing bibliometric networks. This software helps in identifying current interests, research clusters and emerging trends in various subject areas by analyzing bibliographic data. These networks typically comprise scientific research publications, authors, institutions, countries, and keywords. These networks can be constructed based on different parameters, such as co-authorship, citation, co-citation, co-occurrence and bibliographic links.

In this study, several variables were examined, including the publication data, the co-authorship in terms of the number of documents, countries, organizations, citation patterns, and co-citation networks (instances where two documents are cited by the same source) and the co-occurrence of all the keywords. The methodology for this bibliometric analysis is outlined in the accompanying Figure 1.

## 3. Results

### 3.1. Publications and Citations Data

The search performed in the Dimensions database within the period of 2005–2024, after refining the search following the inclusion criteria, resulted in a total of 571 records. As illustrated in Figure 2, the studies on cross-sectional surveys associated with early childhood caries have generally displayed a fluctuating but overall upward trajectory, indicating a growing interest within the academic realm. The peak in publications occurred between 2019 and 2023, with 355 articles published during this period. The graph illustrates the distribution of the publications and the total citations by publication year. Notably, articles published from 2020 to 2023 garnered considerable attention, with the most cited ones being from 2023. This trend in the citation rates may be influenced by the publication year, suggesting that earlier works are more likely to be frequently cited.

### 3.2. Co-Authorship Analysis Regarding Number of Articles

The analysis in this section was centered on the study area of the primary author’s network. Focusing on the two analysis targets (early childhood caries and cross-sectional surveys), 218 authors out of 2346 met the requirement of having at least two published documents per author, 76 met the requirement of having at least three published documents per author, 33 met the requirement of having at least four published documents per author and 23 met the requirement of having at least five published documents per author. The software produced four clusters in total when the threshold was set at five occurrences or more; these clusters were then designated as Groups 1, 2, 3 and 4. Based on the weight of the documents, the author’s network is arranged into four groups, as shown in Figure 3. Table 1 provides further details about the author groups, which indicate that Paiva, Saul Martins authored the most papers (10 documents), with 294 citations and a total link strength of 19. Significant contributions from other writers were also made and will also be included in future studies. The documents that most closely collaborated were found in the red and green groups, followed by Groups 3 (blue) and 4 (yellow).

### 3.3. Co-Authorship Analysis Regarding Countries and Organizations

This section of the present study involved analyzing the scientific co-authorship and investigating the organization of cooperation networks related to the research subject. The behavior of the research groups and their network linkages were identified in this study. The nodes are indicative of either countries or organizations. The distance between the nodes represents the degree of collaboration. Two assessments of scientific co-authorship were carried out: one by country (Figure 4 and Table 2) and the other by organization (Figure 5 and Table 3).

Figure 4 presents 41 countries in nine groups. Our findings show that a vast geographic region was covered in the analysis of the subject of interest, although there were strong connections between various countries around the globe. The first four groups, represented by the red, green, blue and yellow colors, were the most relevant. Australia was the country in the red cluster with the most varied collaborative teams and the strongest associations with other countries (18 TLSs). Other countries, such as Germany (9 documents), Sweden (6 documents), United Arab Emirates (4 documents), the U.K. (17 documents), Brazil (56 documents), Jordan (6 documents), Venezuela (3 documents), Peru (3 documents) and Norway (11 documents), continued their wide-ranging collaboration with other countries. Authors from different countries, like India (68 documents), the United States (52 documents), Indonesia (29 documents), China (28 documents), Iran (24 documents) and Saudi Arabia (23 documents), collaborated more limitedly, despite publishing many articles on the subject.

Similarly, a minimum of four published publications were present in 32 of the 724 organizations, clustered into seven groups, as shown in Figure 5 and Table 3.

### 3.4. Citation Analysis and Co-Citation Network

This analysis concentrated on the citation and network of co-citations of the academic publications, or scientific sources, within our field of study. The journals with the highest number of citations are *BMC Oral Health* with 899 citations and *PLOS One* with 307 citations, as shown in Figure 6. In terms of the co-citation linkages, the relationship between two nodes/sources is shown by their distance from one another. Consequently, a weak link is shown by a wide distance between nodes, and a strong relationship is indicated by a short distance between nodes. The co-citation of scientific publications indicates how frequently one source cited two other sources jointly [20]. The titles of the sources that were taken from the database’s raw reference strings served as the unit of analysis. Only 91 of the 1499 sources fulfilled the requirement of having at least 20 citations per source. As a result, the most prolific journals in our field of study were highlighted in the analysis, which are categorized into six groups, as shown in Figure 7.

The first group, shown in the red color, comprised a total of 33 sources and contained the most cited source, i.e., *BMC Oral Health*, with 447 citations and a total link strength of 10,424. The second group, in the green color, contained journals including *Community Dentistry and Oral Epidemiology* (899 citations and 19,499 total link strength) and *International Journal of Pediatric Dentistry* (510 citations and 12,151 total link strength). The other relevant journal was *Health and Quality of Life Outcomes*, with 185 citations and a total link strength of 4064. The third group, in the blue color, contained relevant sources such as *Pediatrics* (179 citations; 3835 total link strength) and *The Lancet* (132 citations; 2951 total link strength). In our field of study, these journals have made the most substantial contributions overall, making them the most valuable sources.

### 3.5. Keywords Analysis

This analytical approach identifies the most relevant keywords by counting the number of articles that contain a certain keyword at least once. This is because the keywords are the essential information found in an article [21]. For the data set, VOSviewer was used for mapping the network of keywords that appeared at least 10 times in the analyzed articles, like ECC (1147 occurrences), oral health (417 occurrences), preschool child (301 occurrences), OHRQoL (182 occurrences) and Early Childhood Oral Health Impact Scale (ECOHIS) (116 occurrences) etc., implying that preschool children were chosen for cross-sectional surveys regarding ECC. The most relevant keywords, along with the links that connect them, are displayed in Figure 8. A bigger node/keyword denotes more occurrences; a small distance between nodes is indicative of a strong association and an identical color highlights a group or series of related keywords [22]. The outcomes of the cluster mapping suggested that when the threshold was set at 10 occurrences or more, the high-frequency keywords were divided into four clusters. In the network visualization of the keyword co-occurrence, no specific upward or downward trend was seen for any of the terms.

In the present analysis, 19 keywords were determined to be highly relevant (>100 occurrences) based on their relevance order, as depicted in Table 4. It gives the reader a thorough grasp of early childhood caries by offering a structured classification of the keywords in the context of cross-sectional surveys, along with the pertinent tools and major applications [23].

## 4. Discussion

ECC is a multi-factorial and serious concern associated with a greater prevalence and a poor rate of treatment [24]. Children have a polarized distribution of caries, with less than 25% reporting 75% damaged tooth surfaces. Regional and national differences exist in the frequency and severity of ECC. Overall, dental caries in preschool-aged children in South-East Asia is not at an acceptable level. The frequency of caries in these children is around 90% in Laos, Cambodia, and the Philippines. According to the most recent national oral health survey, 70.1% of children (5 y) in China had experienced dental caries [2]. In order to provide important insights into the research patterns, process of development and research hotspots in the field of ECC, a bibliometric and visualization analysis of several cross-sectional surveys was performed in this study using VOSviewer. Our results, which were based on an analysis of 571 open-access articles, indicated that the peak in publications occurred between 2019 and 2023. Cross-sectional studies on ECC have a significant number of citations as well because they show the prevalence and disease linked factors in various groups [25].

Conventional literature reviews are often linked to a certain level of subjectivity and partiality, and thus, they are unable to thoroughly and methodically clarify the aforementioned flaws [26]. Bibliometrics is a document analysis approach that enables the collection and analysis of current research data on a particular topic, both quantitatively and qualitatively. This approach provides a comprehensive overview of the state of the art in cross-sectional surveys related to ECC. Given the scope and diversity of the research, bibliometric analysis can reveal the publishing patterns, historical trends, most prominent authors and emerging topics [27]. This method also makes it possible to identify areas of high consensus and those that need additional research, as well as to evaluate the methodological quality of the included papers in a systematic manner. As a result, such a type of analysis offers a strong basis for developing preventative interventions, public health policies and future research directions [28].

The VOSviewer tool is generally employed for the illustration of bibliometric networks using an automatic algorithm to identify authors as well as terms. The software examines working groups and their associated links, as shown in the co-authorship networks, in addition to the authors yielding high-quality scientific publications. Moreover, it permits the display of the most frequently used terms in the abstract, title, or both, together with the associations between them, i.e., the keyword co-occurrence networks. The terms with a higher correlation are found closer together in the networks, whereas those with a lower correlation are found farther apart [29]. A link is defined as a relationship or connection between two elements in the VOSviewer Manual [30]. For example, in the case of co-authorship networks, the number of documents co-authored by two researchers, or in the case of bibliographic coupling networks, the number of cited references shared by two documents. Each link is represented by a strength value (positive numerical) based on the analysis type [31].

Scientific publications produced by various author groups can be analyzed to help identify and determine the research groups and the relationships that exist within them. The co-authorship of research articles is one way to recognize a collaborative process in science, while there are other ways to communicate scientific collaboration as well. The effectiveness and establishment of scientific partnerships are influenced by the positions of the authors in the co-authorship networks [32]. Despite logical partnerships with different nodes, the produced authorship nodes are clustered under the same institution. The collaborative association between authors and institutions helps in the production of a greater publication count. Researcher scholars as well as professionals will find this information useful as it may be used to identify top researchers as well as organizations in the area that are crucial in spreading knowledge about early childhood caries [33].

The majority of the ECC publications were from researchers in the United States and Brazil. This suggests that nations with stronger economies have a tendency to support biomedical research, maybe as a result of having access to superior financing sources and resources for science and medicine [34]. Nonetheless, a growing number of studies on ECC from developing nations have been published in recent years [35]. The vast majority of the publications in this study came from academic institutions in Brazil and the United States. This is consistent with other bibliometric research conducted in several dental specialties [36].

*BMC Oral Health* is the journal with the highest number of publications as well as citations on this topic, and it is also among the list of 100 journals producing the highest numbers of publications under this aspect [18]. This source includes articles on pathophysiology, molecular genetics, epidemiology, and the diagnosis, treatment and prevention diseases pertaining to oral cavity [37]. The other cited sources, i.e., *Community Dentistry and Oral Epidemiology* and *International Journal of Pediatric Dentistry*, comprise articles on behavioral management, restorative, orthodontic and preventive therapy for children. In general, they encompass all the aspects of community and pediatric dentistry [38]. Likewise, reviews, original research articles and comments related to HRQOL evaluation and therapies are published in other relevant publications. In this study, less than 70% of the articles are open access, which gives them greater visibility and, consequently, a higher citation count from other authors. Most research councils as well as project financing programs now mandate open access to scholarly journals [39].

The categorization of scientific publications is mostly dependent on the keywords. The most substantial knowledge on the direction of study is provided by analyzing the validity and frequency of the keywords [40]. The most relevant keywords in this analysis are not the most commonly occurring ones. The occurrence is represented by the existence, frequency and closeness of the identical keywords. Relevance, on the other hand, shows how significant the word is in the particular field [41]. It should be emphasized that in order to concentrate the information on keywords that will provide the reader with more information, the VOS program chooses 60% of the most pertinent terms that are used in at least 10 publications. With the use of four clusters, each denoted by a distinct color, the primary keyword co-occurrence network provided a thorough overview of the topic of interest within this study’s subject. In the co-occurrence network, the red cluster, which included keywords like “preschool child”, “oral hygiene practice”, “ECC”, “Severe Early Childhood Caries (S-ECC)” and “meta-analysis”, rose to prominence. ECC and oral health were chosen as keywords for screening the database. The green cluster, which contained items like perception, OHRQoL, ECOHIS and oral health, comes next. Other clusters were also found, depicted in blue and yellow, which grouped relevant keywords and represented particular theoretical affinities in the scientific literature on cross-sectional surveys linked to caries in early infancy. The fact that keywords like “dietary habit” (28 occurrences) and “preschool” (49 occurrences) are becoming less common implies that they might not be pertinent to the current work. Although the inclusion of terms like oral hygiene practice (61 occurrences), pediatrician (97 occurrences), and perception (53 occurrences) corelates, the inclusion of behavior (139 occurrences) in the scientific literature, however, suggests that it is becoming increasingly significant and could have an impact on future policy [42].

Bibliometric analyses are inherently limited in a few ways. All open-access publications contained in the Dimensions database served as the only source of data used for the analysis; no other academic research databases were used. The year 2024 was only considered up to 29 May. The search parameters excluded preprints, chapters, and proceedings and only included open-access papers from the limited time frame of 2005 to 2024. This may have resulted in omissions. Second, while the citation count allows for a quantitative assessment of an article’s scientific significance in a certain subject, it does not in itself indicate an article’s quality [43]. Moreover, the potential exclusion of non-English studies or articles in regional journals was also carried out. A few authors also self-cite their works in order to boost the amount of citations, which introduces bias. Lastly, some high-quality articles that have been published recently may have been lost in the enormous ocean of literature data because they have received few citations. The bibliometric study, which was conducted with the use of software, provided an objective viewpoint on the subject matter; however, the interpretation of the findings may have been somewhat subjective.

## 5. Conclusions

The subject of ECC has produced a significant body of scientific literature, which is evolving rapidly. The outcomes of the current study provide an overview of the state of the field’s research on the basis of a bibliometric and visual analysis of cross-sectional studies pertaining to ECC. The bibliometric analytic technique demonstrated the relationships between authors, countries, organizations and sources that are actively engaged in global research as well as the current level of knowledge on the topic under study. This study offered a theoretical framework and recommendations for clinical as well as pre-clinical research on cross-sectional investigations linked to ECC.

## Figures and Tables

**Figure 1 healthcare-13-01067-f001:**
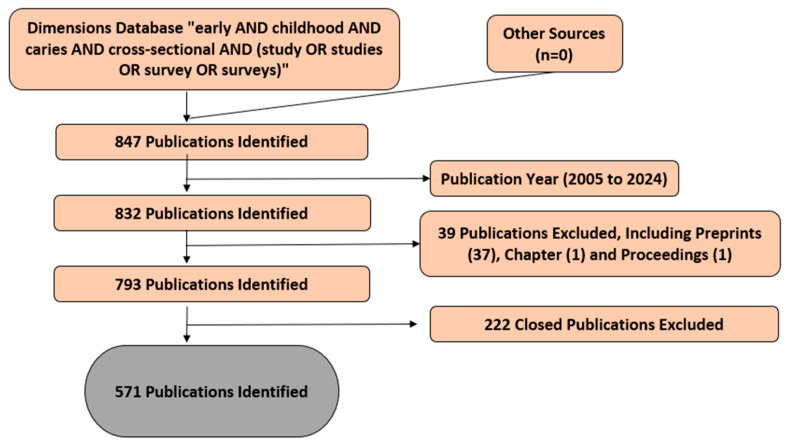
Flow diagram of this bibliometric study.

**Figure 2 healthcare-13-01067-f002:**
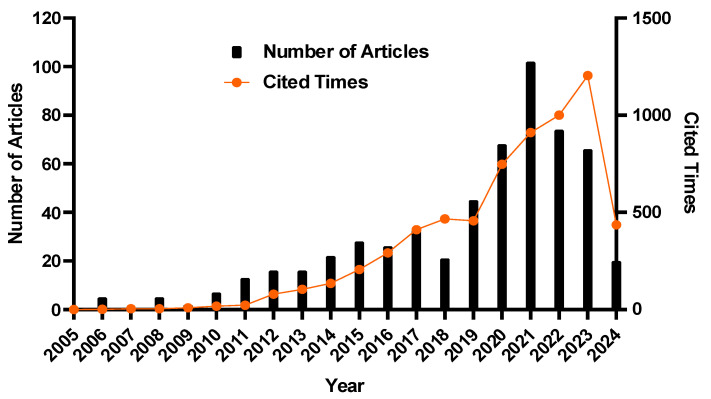
Global patterns of the number of publications and citation frequency from 2005 to 29 May 2024.

**Figure 3 healthcare-13-01067-f003:**
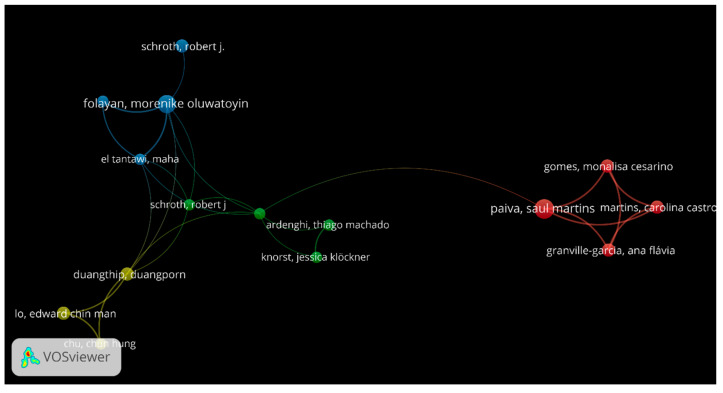
Scientific co-authorship network, based on at least 5 publications per author. The document count is indicated by the bubble size. Collaboration closeness is indicated by the link length.

**Figure 4 healthcare-13-01067-f004:**
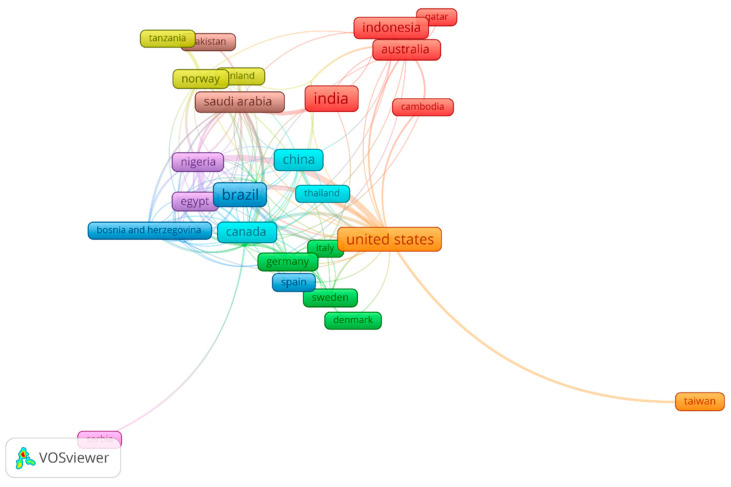
Scientific co-authorship network, based on at least 3 publications per country. The document count is indicated by the frame size. Collaboration closeness is indicated by the link length.

**Figure 5 healthcare-13-01067-f005:**
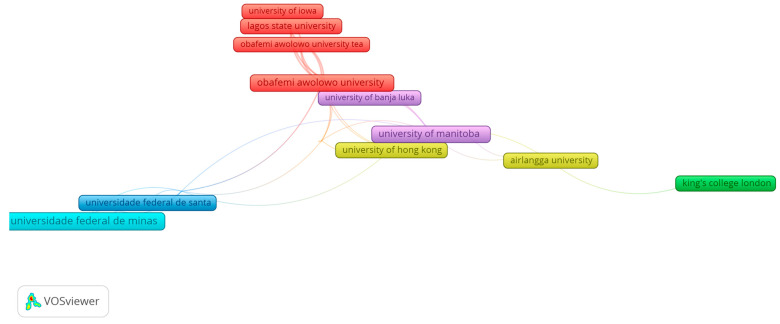
Scientific co-authorship network, based on at least 4 publications per organization. The document count is indicated by the frame size. Collaboration closeness is indicated by the link length.

**Figure 6 healthcare-13-01067-f006:**
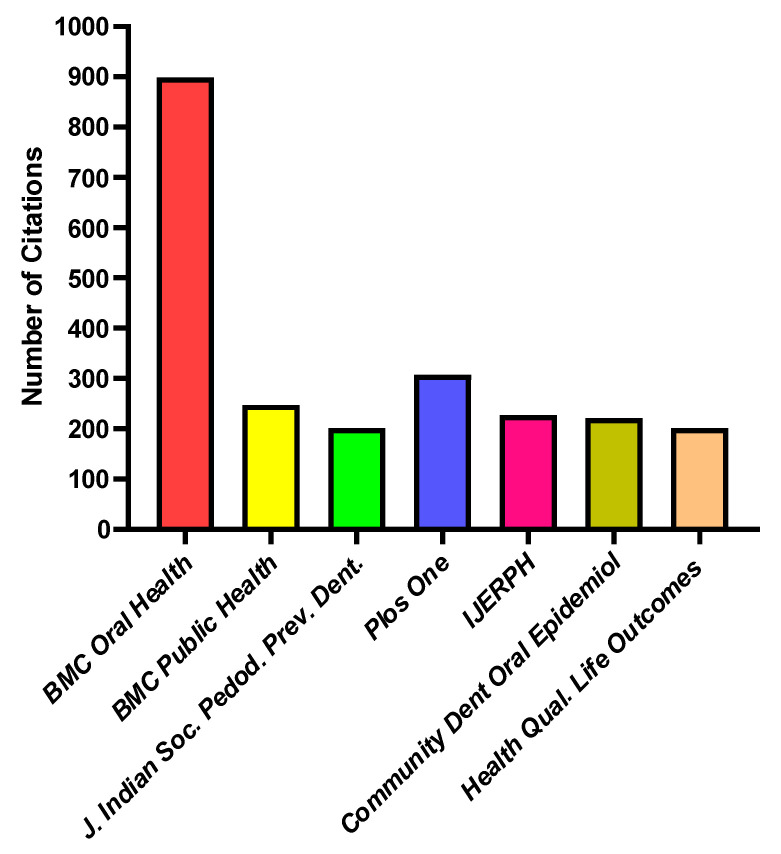
Journals with at least 5 citations with the topic “early AND childhood AND caries AND cross-sectional AND (study OR studies OR survey OR surveys)”.

**Figure 7 healthcare-13-01067-f007:**
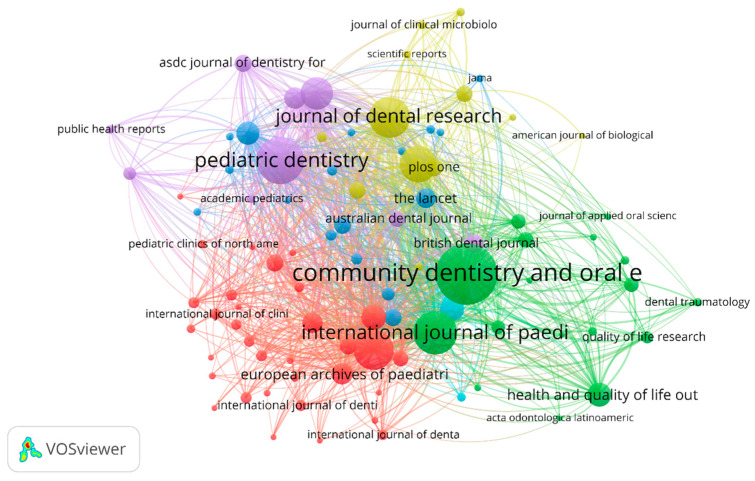
Scientific co-citation network, based on at least 20 citations per source. The number of citations is indicated by the bubble size. Collaboration closeness is indicated by the link length. *BMC Oral Health* = 447 citations and 10424 TLS.

**Figure 8 healthcare-13-01067-f008:**
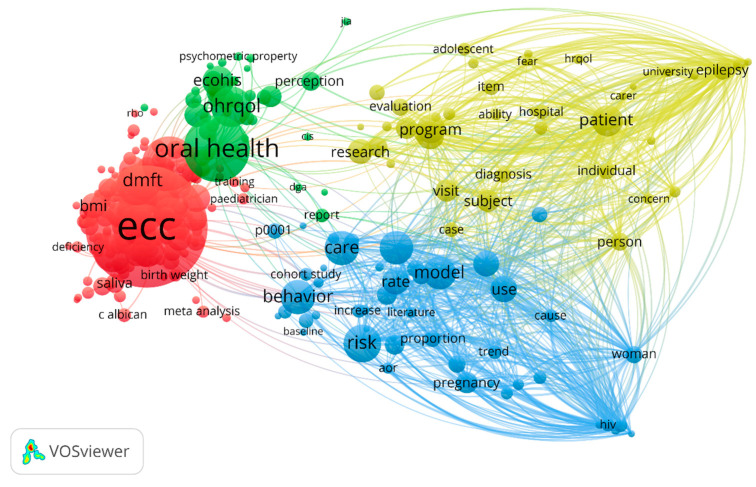
Network visualization of keyword co-occurrence (minimum 10 occurrences). The bubble size represents the occurrence frequency.

**Table 1 healthcare-13-01067-t001:** Author groups and related document counts, citations and total link strengths (TLSs).

Groups	Documents	Percentage (%)	Citations	ACI	TLS
Group 1 (Red)					
Paiva, Saul Martins	10	1.8	294	29.4	19
Gomes, Monalisa Cesarino	6	1.1	280	46.7	18
Granville-Garcia, Ana Flavia	6	1.1	280	46.7	18
Martins, Carolina Castro	6	1.1	280	46.7	18
Group 2 (Green)					
Feldens, Carlos Alberto	5	0.9	197	39.4	7
Ardenghi, Thiago Machado	5	0.9	22	4.4	4
Knorst, Jessica Klockner	5	0.9	20	4	4
Schroth, Robert j	5	0.9	199	39.8	4
Group 3 (Blue)					
Folayan, Morenike Oluwatoyin	9	1.6	108	12	14
El Tantawi, Maha	5	0.9	74	14.8	11
Schroth, Robert j.	6	1.1	193	32.2	1
Finlayson, Tracy I.	5	0.9	79	15.8	8
Group 4 (Yellow)					
Duangthip, Duangporn	6	1.1	83	13.8	12
Chu, Chun hung	5	0.9	59	11.8	9
Lo, Edward Chin Man	6	1.1	60	10	9

ACI = Average citations per item, TLS = Total link strength.

**Table 2 healthcare-13-01067-t002:** Published documents, citations and total link strengths grouped by country.

Groups	Documents and Percentage (%)	Citations and ACI	TLS	Groups	Documents and Percentage (%)	Citations and ACI	TLS
Group 1				Group 2			
Australia	18 (3.2)	178 (9.9)	18	Denmark	3 (0.5)	49 (16.3)	3
Cambodia	3 (0.5)	26 (8.7)	5	France	3 (0.5)	62 (20.6)	6
India	68 (11.9)	690 (10.1)	9	Germany	9 (1.6)	215 (23.9)	12
Indonesia	29 (5.1)	46 (1.6)	5	Italy	4 (0.7)	109 (27.3)	7
Malaysia	8 (1.4)	52 (6.5)	4	Netherlands	3 (0.5)	113 (37.7)	3
New Zealand	5 (0.9)	70 (14.0)	7	Sweden	6 (1.1)	38 (6.3)	11
Qatar	3 (0.5)	60 (20.0)	1	U.A.E.	4 (0.7)	82 (21.0)	12
Sri Lanka	3 (0.5)	70 (23.3)	1	U.K.	17 (3.0)	365 (21.5)	31
Group 3				Group 4			
Bosnia & Herzegovina	5 (0.9)	19 (3.8)	9	Finland	4 (0.7)	44 (11)	7
Brazil	56 (9.8)	1280 (22.9)	21	Iran	24 (4.2)	318 (13.3)	6
Chile	8 (1.4)	83 (10.4)	6	Norway	11 (1.9)	206 (18.7)	22
Jordan	6 (1.1)	50 (8.3)	21	Tanzania	5 (0.9)	174 (34.8)	4
Kenya	3 (0.5)	26 (8.7)	9	Group 5			
Peru	3 (0.5)	7 (2.3)	12	Egypt	13 (2.3)	97 (7.5)	33
Spain	6 (1.1)	60 (10)	4	Nigeria	16 (2.8)	212 (13.3)	39
Venezuela	3 (0.5)	17 (5.7)	13	South Africa	9 (1.6)	38 (4.2)	14
Group 6				Group 7			
Canada	20 (3.5)	740 (37)	33	Taiwan	7 (1.2)	70 (10)	4
China	28 (4.9)	468 (16.7)	15	U.S.	52 (9.1)	1325 (25.5)	50
Thailand	5 (0.9)	67 (13.4)	3	Vietnam	5 (0.9)	58 (11.6)	1
Group 8				Group 9			
Pakistan	4 (0.7)	19 (4.8)	2	Serbia	4 (0.7)	25 (6.3)	3
Saudi Arabia	23 (4.0)	254 (11.0)	31	Switzerland	3 (0.5)	43 (14.3)	1

ACI = Average citations per item, TLS = Total link strength.

**Table 3 healthcare-13-01067-t003:** Published documents, citations and total link strengths grouped by organization.

Groups	Documents and Percentage (%)	Citations and ACI	TLS
Group 1			
Obafemi Awolowo University	12 (2.1)	179 (14.9)	32
Lagos State University	7 (1.2)	86 (12.3)	21
San Diego State University	5 (0.9)	79 (15.8)	19
Obafemi Awolowo University Tea	6 (1.1)	169 (28.2)	18
Alexandria University	8 (1.4)	85 (10.6)	28
University of Iowa	4 (0.7)	42 (10.5)	2
Group 2			
King’s College London	7 (1.2)	158 (22.6)	3
Ministry of Health	6 (1.1)	30 (5)	5
Peking University	6 (1.1)	159 (26.5)	1
Qassim University	4 (0.7)	23 (5.8)	1
University of la Frontera	5 (0.9)	51 (10.2)	1
University of Melbourne	7 (1.2)	82 (11.7)	4
Group 3			
Federal University of Rio Grande do Sul	5 (0.9)	41 (8.2)	3
State University of Campinas	6 (1.1)	107 (17.8)	1
Universidade de Sao Paulo	6 (1.1)	109 (18.2)	4
Universidade Federal de Santa Maria	8 (1.4)	91 (11.4)	6
Universidade Luterana do Brasil	7 (1.2)	200 (28.6)	12
University of California, San Francisco	4 (0.7)	205 (51.3)	3
Group 4			
Airlangga University	9 (1.6)	21 (2.3)	2
University of Adelaide	4 (0.7)	55 (13.8)	4
University of Hong Kong	10 (1.8)	155 (15.5)	6
University of Otago	5 (0.9)	70 (14)	6
University of Toronto	4 (0.7)	315 (78.8)	4
Group 5			
Children’s Hospital Research Institute of Manitoba	6 (1.1)	288 (48)	9
Tehran University of Medical Sciences	4 (0.7)	17 (4.3)	1
University of Banja Luka	5 (0.9)	19 (3.8)	3
University of Manitoba	13 (2.3)	403 (31)	20
Winnipeg Regional Health Authority	4 (0.7)	240 (60)	8
Group 6			
State University of Paraiba	7 (1.2)	302 (43.1)	7
Universidade Federal de Minas Gerais	15 (2.6)	604 (40.3)	9
Group 7			
Muhimbili University of Health and Allied Sciences	5 (0.9)	174 (34.8)	4
University of Bergen	9 (1.6)	191 (21.2)	11

ACI = Average citations per item, TLS = Total link strength.

**Table 4 healthcare-13-01067-t004:** Highly relevant keywords (>100 occurrences) in the order of relevance.

Serial No.	Term	Occurrences	Relevance
1	ECOHIS	116	1.51
2	S ECC	156	1.17
3	OHRQoL	182	1.04
4	Bottle	128	1.03
5	Habit	124	0.96
6	DMFT	168	0.94
7	Early Childhood Caries	107	0.94
8	Preschool Child	301	0.92
9	Childhood Caries	130	0.91
10	Index	181	0.90
11	ECC	1147	0.85
12	Patient	119	0.81
13	Oral Health	417	0.73
14	Program	103	0.52
15	Risk	164	0.47
16	Model	130	0.30
17	Behavior	139	0.29
18	Population	137	0.20
19	Care	140	0.15

## Data Availability

The original contributions presented in this study are included in the article. Further inquiries can be directed to the corresponding author.

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
