# Peer review of "Bibliometric Analysis of Cross-Sectional Studies on Early Childhood Caries"

_healthcare, 2025, doi:10.3390/healthcare13091067_

Round 1

Reviewer 1 Report

Comments and Suggestions for Authors

Dear Authors,

The manuscript presents a bibliometric analysis of cross-sectional surveys on early childhood caries (ECC), offering insights into research trends, collaborations, and keyword networks. While the study addresses a relevant gap in the literature, several improvements are needed to enhance clarity, methodological transparency, and scientific rigor. Below are detailed comments organized by section.

Title (Page 1, Line 1)

  • The title is descriptive but lengthy. Consider simplifying it for conciseness while retaining key elements, e.g., "Bibliometric Analysis of Cross-Sectional Studies on Early Childhood Caries."

Abstract

  • The phrase "inconsistencies and gaps remain" is vague. Specify the nature of the gaps (e.g., geographic disparities, methodological limitations) to strengthen the rationale.
  • The term "total link strength" is introduced without definition. Consider adding a brief explanation (e.g., "a measure of collaborative ties") for non-specialist readers.

Introduction

  • Repetitive statements about ECC’s impact on quality of life. Condense to avoid redundancy.
  • The claim of ECC prevalence reaching "as high as 90% in developing countries" is striking but lacks specificity (e.g., which countries or studies?). This reduces its scientific weight.
  • The justification for bibliometric analysis lacks specificity. Clarify how this study differs from prior reviews (e.g., Vega et al., Patil et al.) beyond "focusing on cross-sectional surveys."

Methods

  • The cutoff year (2024) introduces incomplete data, as the search was conducted mid-2024. Acknowledge this limitation in the discussion.
  • The rationale for VOSviewer thresholds (e.g., ≥5 documents per author) is unclear. Explain how these thresholds balance inclusivity and analytical focus.
  • The flow diagram lacks clarity. Label axes in Figure 2 (e.g., "Year" should specify 2005–2023) and define abbreviations (e.g., "TLS").

Results

  • Tables 1–3
    • Group labels (e.g., "Red," "Green"): Define cluster colors in figure captions (e.g., Figure 3) to avoid forcing readers to cross-reference text.
    • Table 2: The term "ACI" (Average Citations per Item) is undefined. Add a footnote or expand the abbreviation at first mention.
  • Figures 3–5
    • Node/link interpretations (e.g., "distance = collaboration closeness") require explicit definitions in captions.
    • Figure 8: The keyword relevance metric (Table 4) is unexplained. Clarify how "relevance" is calculated (e.g., VOSviewer’s relevance score vs. manual scoring).

Discussion

  • The statement about nations with "stronger economies" supporting research is speculative. Provide citations to support this claim or rephrase as a hypothesis.
  • The limitation regarding database bias (Dimensions-only data) is underemphasized. Expand to discuss the potential exclusion of non-English studies or articles in regional journals.
  • The discussion of keyword trends (e.g., "dietary habit" declining) lacks contextualization. Link findings to shifts in research priorities (e.g., rising focus on behavioral factors).

Conclusions

  • The conclusion overstates impact with phrases like "comprehensively examined." Replace with measured language (e.g., "provides an overview").

References

  • Multiple references lack consistent formatting (e.g., journal names italicized inconsistently, missing DOIs). Standardize per journal guidelines.
Comments on the Quality of English Language

Polish the writing for conciseness and consistency, ensuring all references and figures are complete.

Reviewer 2 Report

Comments and Suggestions for Authors

Dear Author,

Thank you for submitting the manuscript. You have implemented an exciting idea and chosen an interesting way to present the topic of ‘ECC’.

However, a few points should be revised so that the reader can better understand your methodology and the results found.

In general

You speak of ‘our study’ and ‘we searched’ etc.. However, only one person is named as the author, and only one person is named under ‘Author Contributions’. Either do not use the plural in the manuscript or name the number of people involved in the material and methods (Abbreviation) or, if necessary, name co-authors.

M&M

Could you name the ‘dimension database’ used and explain it in more detail? Unfortunately, this reads as unclear and is not comprehensible to the reader.

Was a validation of the examiners performed?

Could you explain the aspects under which you considered studies to be irrelevant and therefore excluded them?

Could you specify the specific program versions of Excel and VoS Viewer (manufacturer, location)?

Could you explain the Vos Viewer program in more detail?

Could you explain the procedures ‘Co-Authorship Analysis Regarding’ and ‘Citation Analysis & Co-citation Network’ in M&M in more detail? Unfortunately, this is not yet sufficiently comprehensible.

Results:

The statement that 2023 was the highest publication should be put into perspective, since you could only evaluate the year 2024 up to your cut-off date (29 May 2024). You could therefore state in your Figure 2 that the year 2024 was only considered up to 29 May.

Figure 3 and Table 1 I believe that the first names and surnames are in the wrong order here. Could you check and correct this?

Figure 7: If I understand this correctly, the journal BMC Oral Health is the most frequently cited? But it is not mentioned in Figure 7?

3.5 Keyword analysis

I find the analysis somewhat flawed. Surely it must be assumed that the keywords ‘ECC’ and ‘oral health’ are used if this is the content of the manuscript? Could you include this point in the discussion?

Table 4: Could you explain again how you calculated the ‘relevance’? That is, why is ‘ECOHIS’ more relevant than ‘ECC’ (1147 occurrences), even though this term was used more frequently?

Discussion:

Many of the points you list could also be listed under the M&M section.

For example, I would highly recommend that you compare your methodology with other studies. In your introduction, you mention several articles (e.g. Patil, Melo) that you would like to take up and reflect on in your work.

Round 2

Reviewer 2 Report

Comments and Suggestions for Authors

Dear Author,

Thank you very much for your revision.

Unfortunately, not all of the changes you made are highlighted in colour in your document, so I may have duplicated some of my comments regarding improvements. Please accept my apologies for this.

Firstly, please specify the version, manufacturer and location/country under ‘(Microsoft 365)’. Please do the same for the ‘VOS’ software.

‘Any discrepancies were resolved through discussions involving a third researcher (examiner validation), culminating in a consensus.’ Could you explain exactly how you did this? How many examiners were involved in the study and what were their specific tasks?

Figure 3 and Table 1 I believe that the first names and surnames are in the wrong order here. Could you check and correct this? Knorst Klöckner, Jessica/ Martins Castro, Carolina

Figure 5. Unfortunately, the figure is not completely legible. One area is not fully displayed.

Figure 7: Thank you for the correction. Could BMC also display this within the graph?

3.5 Keyword analysis
I find the analysis somewhat flawed. Surely it must be assumed that the keywords ‘ECC’ and ‘oral health’ are used if this is the content of the manuscript? Could you include this point in the discussion?

--> Could you please explain this in more detail? Thank you very much.

Table 4: Could you explain again how you calculated the ‘relevance’? That is, why is ‘ECOHIS’
more relevant than ‘ECC’ (1147 occurrences), even though this term was used more frequently?
Answer: Relevance calculation was provided by VOSviewer software.
--> Could you explain this in the manuscript? Which algorithm did the VOS software use to display this?

Discussion:
Many of the points you list could also be listed under the M&M section.
For example, I would highly recommend that you compare your methodology with other studies. In your introduction, you mention several articles (e.g. Patil, Melo) that you would like to take up and reflect on in your work.

Answer: Our bibliometric analysis utilizes Dimensions database for data extraction whereas
earlier reviews focused on Scopus and Web of Science for extraction of data.

--> Would you be able to compare your methodology with other studies in this publication?

Author Response

Firstly, please specify the version, manufacturer and location/country under ‘(Microsoft 365)’. Please do the same for the ‘VOS’ software.

Answer: Thank you for your valuable comments, it was addressed as suggested. Highlighted in yellow color.

‘Any discrepancies were resolved through discussions involving a third researcher (examiner validation), culminating in a consensus.’ Could you explain exactly how you did this? How many examiners were involved in the study and what were their specific tasks?

Answer: Thank you for your valuable comments, they were addressed as suggested. Highlighted in yellow color.

Figure 3 and Table 1 I believe that the first names and surnames are in the wrong order here. Could you check and correct this? Knorst Klöckner, Jessica/ Martins Castro, Carolina

Answer: Already mentioned in the first rebuttal. The dimension software extract and provide the required tables so the name orders are given by software. We cannot modify it.

Figure 5. Unfortunately, the figure is not completely legible. One area is not fully displayed.

Answer: Thank you for your valuable comments, it was addressed as suggested.

Figure 7: Thank you for the correction. Could BMC also display this within the graph?

Answer: The dimension software extract and provide the required graphs. We cannot modify it. That’s why it is mentioned in the legend.

3.5 Keyword analysis
I find the analysis somewhat flawed. Surely it must be assumed that the keywords ‘ECC’ and ‘oral health’ are used if this is the content of the manuscript? Could you include this point in the discussion?

--> Could you please explain this in more detail? Thank you very much.

Answer: Thank you for your valuable comments, it was addressed as suggested. Highlighted in yellow color.

Table 4: Could you explain again how you calculated the ‘relevance’? That is, why is ‘ECOHIS’
more relevant than ‘ECC’ (1147 occurrences), even though this term was used more frequently?
Answer: Relevance calculation was provided by VOSviewer software.
--> Could you explain this in the manuscript? Which algorithm did the VOS software use to display this?

Answer: Thank you for your valuable comments. The software has preset parameters that automatically provide the required values after putting in place the data in the software. Therefore, this relevance value is automatically generated by software.

Discussion:
Many of the points you list could also be listed under the M&M section.
For example, I would highly recommend that you compare your methodology with other studies. In your introduction, you mention several articles (e.g. Patil, Melo) that you would like to take up and reflect on in your work.

Answer: Our bibliometric analysis utilizes Dimensions database for data extraction whereas
earlier reviews focused on Scopus and Web of Science for extraction of data.

--> Would you be able to compare your methodology with other studies in this publication?

Answer: Thank you for your valuable comments. Since we are dealing with bibliometric analysis according to our own predetermined methodology, there is no comparison with the previously reported bibliometric analysis manuscripts as each and every software provide its own data without any comparison.
